# Crossing the Aisle: Unveiling Partisan and Counter-Partisan Events in News Reporting

**Kaijian Zou**[1]   **Xinliang Frederick Zhang**[1]
**Winston Wu**[2]   **Nick Beauchamp**[3]   **Lu Wang**[1]

[1]Computer Science and Engineering, University of Michigan, Ann Arbor, MI
[2]Department of Computer Science, University of Hawaii at Hilo, Hilo, HI
[3]Department of Political Science, Northeastern University, Boston, MA
[1]{zkjzou,xlfzhang,wangluxy}@umich.edu
[2]wswu@hawaii.edu, [3]n.beauchamp@northeastern.edu

## Abstract

News media is expected to uphold unbiased reporting. Yet they may still affect public opinion by selectively including or omitting events that support or contradict their ideological positions. Prior work in NLP has only studied media bias via linguistic style and word usage. In this paper, we study to which degree media balances news reporting and affects consumers through event inclusion or omission. We first introduce the task of detecting both **partisan** and **counter-partisan** events: events that support or oppose the author's political ideology. To conduct our study, we annotate a high-quality dataset, PAC, containing $8,511$ (counter-)partisan event annotations in 304 news articles from ideologically diverse media outlets. We benchmark PAC to highlight the challenges of this task. Our findings highlight both the ways in which the news subtly shapes opinion and the need for large language models that better understand events within a broader context. Our dataset can be found at https://github.com/launchnlp/Partisan-Event-Dataset.

## 1 Introduction

Political opinion and behavior are significantly affected by the news that individuals consume. There is now extensive literature examining how journalists and media outlets promote their ideologies via moral or political language, tone, or issue framing (de Vreese, 2004; DellaVigna and Gentzkow, 2009; Shen et al., 2014; Perse and Lambe, 2016). However, in addition to this more overt and superficial *presentation bias*, even neutrally written, broadly-framed news reporting, which appears both "objective" and relatively moderate, may shape public opinion through a more invisible process of *selection bias*, where factual elements that are included or omitted themselves have ideological effects (Gentzkow and Shapiro, 2006; D'Alessio and

**Story Title:** Texas Governor Signs 'Heartbeat Bill' Banning Abortion

**National Review** (Right):
Texas Governor Greg Abbott [signed] a bill on Wednesday barring abortions · · · "Our creator endowed us with the right to life and yet millions of children [lose] their right to life every year because of abortion," Abbott, a Republican, said during a bill signing ceremony. · · ·

**Event #1** ( [sign] ): pos->*Heartbeat Bill*
**Event #2** ( [lose] ): pos->*Heartbeat Bill*   neg->*abortion*

**The Guardian** (Left):
The Texas Republican governor Greg Abbott has [signed] into law one of the most extreme six-week abortion bans in the US, despite strong [opposition] from the medical and legal communities · · ·."This bill ensures that every unborn child who has a heartbeat will be [saved] from the ravages of abortion," said Abbott · · ·.

**Event #3** ( [signed] ): neg->*Heartbeat Bill*
**Event #4** ( [opposition] ): neg->*Heartbeat Bill*
**Event #2** ( [saved] ): pos->*Heartbeat Bill*   neg->*abortion*

Figure 1: Excerpt from a news story reporting on Heartbeat Bill. [Blue] indicate events favoring left-leaning entities and disfavoring right-leaning entities; vice versa for [Red]. Although both media outlets report the event Greg Abbott **signed** Heartbeat Bill, The Guardian select the additional event **opposition** to attack Heartbeat Bill. Interestingly, both media outlets include quotes from Greg Abbott but for different purposes: one for supporting the bill and the other for balanced reporting.

Allen, 2006; Groeling, 2013).

Existing work in NLP has only studied bias at the token- or sentence-level, particularly examining how language is phrased (Greene and Resnik, 2009; Yano et al., 2010; Recasens et al., 2013; Lim et al., 2020; Spinde et al., 2021). This type of bias does not rely on the context outside of any individual sentence, and can be altered simply by using different words and sentence structures. Only a few studies have focused on bias that depends on broader contexts within a news article (Fan et al.,

2019; van den Berg and Markert, 2020) or across articles on the same newsworthy event (Liu et al., 2022; Qiu et al., 2022). However, these studies are limited to token- or span-level bias, which is less structured, and fail to consider the more complex interactions among news entities.

To understand more complex content selection and organization within news articles, we scrutinize how media outlets include and organize the fundamental unit of news–**events**–to subtly reflect their ideology while maintaining a seemingly balanced reporting. Events are the foundational unit in the storytelling process (Schank and Abelson, 1977), and the way they are selected and arranged affects how the audience perceives the news story (Shen et al., 2014; Entman, 2007). Inspired by previous work on selection bias and presentation bias (Groeling, 2013; D'Alessio and Allen, 2006), we study two types of events. (i) **Partisan events**, which we define as *events that are purposefully included to advance the media outlet's ideological allies' interests or suppress the beliefs of its ideological enemies*. (ii) **Counter-partisan events**, which we define as *events purposefully included to mitigate the intended bias or create a story acceptable to the media industry's market*. Figure 1 shows examples of partisan events and counter-partisan events.

To support our study, we first collect and label **PAC**, a dataset of 8,511 PArtisan and Counter-partisan events in 304 news articles. Focusing on the partisan nature of media, PAC is built from 152 sets of news stories, each containing two articles with distinct ideologies. Analysis on PAC reveals that partisan entities tend to receive more positive sentiments and vice versa for count-partisan entities. We further propose and test three hypotheses to explain the inclusion of counter-partisan events, considering factors of newsworthiness, market breadth, and emotional engagement.

We then investigate the challenges of partisan event detection by experimenting on PAC. Results show that even using carefully constructed prompts with demonstrations, ChatGPT performs only better than a random baseline, demonstrating the difficulty of the task and suggesting future directions on enabling models to better understand the broader context of the news stories.

## 2   Related Work

Prior work has studied media bias primarily at the word-level (Greene and Resnik, 2009; Recasens et al., 2013) and sentence-level (Yano et al., 2010; Lim et al., 2020; Spinde et al., 2021). Similar to our work is informational bias (Fan et al., 2019), which is defined as "tangential, speculative, or background information that tries to sway readers' opinions towards entities in the news." However, they focus on span-level bias, which does not necessarily contain any salient events. In contrast, our work considers bias on the event level, which is neither "tangential" to news, nor at the token level. Importantly, we examine both *partisan* and *counter-partisan* events in order to study how these core, higher-level units produce ideological effects while maintaining an appearance of objectivity.

Our work is also in line with a broad range of research on *framing* (Entman, 1993; Card et al., 2015), in which news media select and emphasize some aspects of a subject to promote a particular interpretation of the subject. Partisan events should be considered as *one type of framing* that focuses on fine-grained content selection phenomenon, as writers include and present specific "facts" to support their preferred ideology. Moreover, our work relates to research on the selection or omission of news items that explicitly favor one party over the other (Entman, 2007; Gentzkow and Shapiro, 2006; Prat and Strömberg, 2013), or selection for items that create more memorable stories (Mullainathan and Shleifer, 2005; van Dalen, 2012). In contrast, we focus on core news events, those that may not explicitly favor a side, but which are nevertheless ideological in their effect.

Finally, our research is most similar to another recent study on partisan event detection (Liu et al., 2023), but they only investigate *partisan events* and focus on developing computational tools to detect such events. In contrast, our work also incorporates *counter-partisan events*, enabling a broader and deeper understanding of how media tries to balance impartial news coverage and promoting their own stances. We also construct a significantly larger dataset than the evaluation set curated in Liu et al. (2023), enhancing its utility for model training.

## 3   Partisan Event Annotation

PAC contains articles from two sources. We first sample 57 sets of news stories published between 2012–2022 from SEESAW (Zhang et al., 2022). Each news story set contains three articles on the same story from outlets with different ideologies. Here we take out the articles labeled with a Cen-

|  | Train | Dev | Test | All |
|---|---|---|---|---|
| # of stories | 95 | 22 | 35 | 152 |
| # of articles | 190 | 44 | 70 | 304 |
| # of sentences | 5,206 | 1,410 | 1,727 | 8,343 |
| # of events | 15,947 | 4,301 | 5,628 | 25,876 |
| # of partisan events | 4,131 | 924 | 1,377 | 6,432 |
| # of counter-partisan events | 1,357 | 280 | 442 | 2,079 |
| Time range | 2012-2021 | 2022-2022.06 | 2022.07-2023 | 2012-2023 |

Table 1: Statistics of PAC.

ter ideology and only keep stories with two news articles from opposite ideologies. To increase the diversity of topics in our dataset, we further collect 95 sets of news stories from www.allsides.com, covering topics such as abortion, gun control, climate change, etc. We manually inspect each story and keep the ones where the two articles are labeled with left and right ideologies. Next, we follow the definition of events from TimeML (Pustejovsky et al., 2003), i.e., a cover term for situations that happen or occur, and train a RoBERTa-Large model on MATRES (Ning et al., 2018) for event detection. Our event detector achieves an F1 score of 89.31, which is run on PAC to extract events.

Next, the partisan events are annotated based on the following process. For each pair of articles in a story, an annotator is asked to first read both articles to get a balanced view of the story. Then, at the article level, the annotator determines the **relative ideological** ordering, i.e., which article falls more on the left (and the other article more right) on the political spectrum. Then, the annotator estimates each article's **absolute ideology** on a 5-point scale, with 1 being far left and 5 as far right.

For each event in an article, annotators first identify its **participating entities**, i.e., who enables the action and who is affected by the events, and assign them an **entity ideology** when appropriate, and estimate the **sentiments** they receive if any. Using the story context, the article's ideology, and the information of the participating entities, annotators label each event as **partisan**, **counter-partisan**, or **neutral** relative to the article's ideology, based on the definitions given in the introduction. If an event is labeled as non-neutral, annotators further mark its intensity to indicate how strongly the event supports the article's ideology. A complete annotation guideline can be found in Appendix A. The annotation quality control process and inter-annotator agreement are described in Appendix B. We also discuss disagreement resolution in Appendix C. The final dataset statistics are listed in Table 1.

## 4 Descriptive Analysis

**Partisan event selection effect.** As shown in Figure 2, more ideologically extreme outlets include

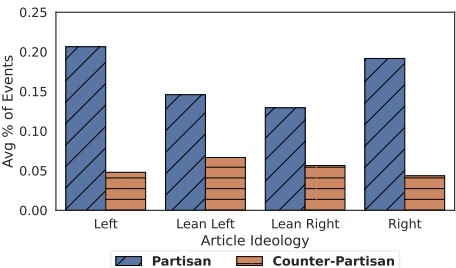

Figure 2: Average percentage of partisan and counter-partisan events reported across articles for media with different ideologies. More moderate news outlets tend to include a more equal mix of events.

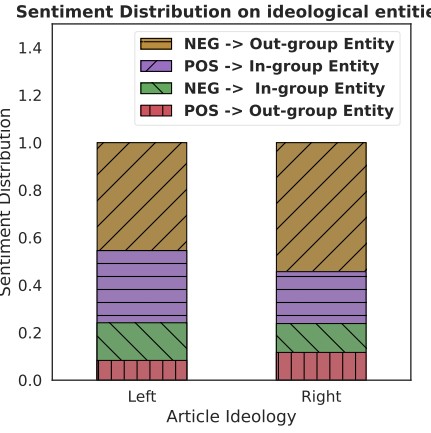

Figure 3: Distribution of positive and negative portray of left and right entities. News events tend to be positive toward in-group entities and negative toward out-groups.

many more partisan events than counter-partisan events, whereas more moderate news outlets tend to include a more equal mix.

**Partisan sentiment.** News media also reveal their ideology in the partisan entities they discuss, via the sentiments associated with those entities, where partisan entities tend to have positive associations and vice versa for count-partisan entities (Groeling, 2013; Zhang et al., 2022). In Figure 3, we find support for this expectation. We also find that left entities generally receive more exposure in articles from both sides.

**Partisan event placement.** Figure 4 shows that for both left and right media outlets, partisan events appear a bit earlier in news articles. For counter-partisan events, left-leaning articles also place more counter-partisan events at the beginning, while right-leaning articles place more counter-partisan events towards the end. This asymmetry suggests that right-leaning outlets are more sensitive to driving away readers with counter-partisan events, thus placing them at the end of articles to avoid that.

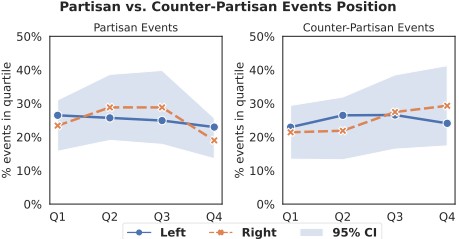

Figure 4: Distribution of partisan and counter-partisan events in each quartile per news article. Shaded area shows 95% confidence level for both left and right.

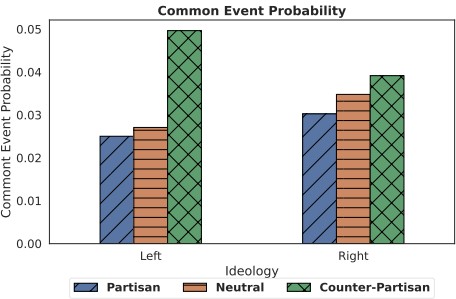

Figure 5: The probability of each type of event being common in left and right articles. If an event is counter-partisan, it will more likely be a common event.

# 5 Explaining Partisan and Counter-Partisan Event Usage

In this section, we investigate a number of hypotheses about why media outlets include both partisan and counter-partisan events. It is intuitive to understand why partisan events are incorporated into the news storytelling processing, yet it is unclear why counter-partisan events that portray members of one's own group negatively or members of another group favorably are reported. Specifically, we establish and test three hypotheses for why an outlet would include counter-partisan news, similar to some of the theories articulated in Groeling (2013): (1) newsworthiness, (2) market breadth, and (3) emotional engagement.

## 5.1 Hypothesis 1: Newsworthiness

This hypothesis suggests that a primary goal of mainstream media is to report newsworthy content, even if it is counter-partisan. In Figure 5, we find that counter-partisan events are more likely to be reported by both sides (which is not tautological because the ideology of events is not simply inferred from article ideology). However, we find a striking asymmetry, where the left appears to report mainly counter-partisan events that were also reported on by the right, but the counter-partisan events reported by the right are not as common on the left. This suggests that the left may be motivated by newsworthiness more.

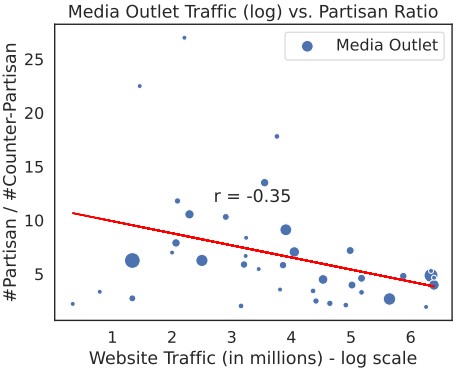

Figure 6: The average ratio of partisan vs. counter-partisan events by media outlets versus logged website traffic. The size of dots represents media's article number in PAC. Larger media tends to be more balanced.

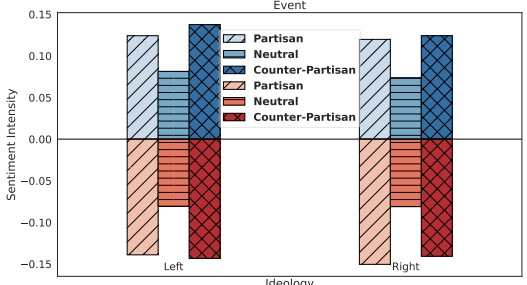

Figure 7: The average sentiment intensity of each type of event measured by VADER. Blue indicates positive sentiments, and red indicates negative sentiments. Both partisan and counter-partisan are associated with stronger sentiments.

## 5.2 Hypothesis 2: Market Breadth

Our second hypothesis is that media may seek to preserve a reputation of moderation, potentially in order not to drive away a large segment of its potential audience (Hamilton, 2006). One implication of this hypothesis is that larger media either grew through putting this into practice, or seek to maintain their size by not losing audience, while smaller media can focus on more narrowly partisan audiences. To test this implication, we collected the monthly website traffic [1] of each media outlet with more than one news article in our dataset and computed the average ratio of partisan to counter-partisan events, calculated per article and then averaged over each outlet. In Figure 6, we plot the average partisan ratio against the logged monthly website traffic. The correlation coefficient of -0.35 supports the hypothesis that larger outlets produce a more bipartisan account of news stories.

## 5.3 Hypothesis 3: Emotional Engagement

Our third hypothesis is that outlets will include counter-partisan content if its benefits in terms

---

[1] We collected data from www.similarweb.com.

of emotional audience engagement outweigh its ideological costs (Gentzkow and Shapiro, 2006). This implies that the emotional intensity of counter-partisan events should be higher than that of partisan events (since higher intensity is required to off-set ideological costs). We employ VADER (Hutto and Gilbert, 2014), a lexicon and rule-based sentiment analysis tool on each event to compute its sentiment intensity. Figure 7 shows that both partisan and counter-partisan events have stronger sentiments than non-partisan events, but we find no strong support for our hypothesis that counter-partisan events will be strongest. If anything, right-leaning events are more intense when reported on by either left or right media, but this difference is not statistically significant.

## 6 Experiments

We experiment on PAC for two tasks. **Partisan Event Detection**: Given all events in a news article, classify an event as partisan, counter-partisan, or neutral. **Ideology Prediction**: Predict the political leaning of a news article into left or right.

### 6.1 Models

We experiment with the following models for the two tasks. We first compare with a **random** baseline, which assigns an article's ideology and an event's partisan class based on their distribution in the training set. Next, we compare to **RoBERTa**-base (Liu et al., 2019) and **POLITICS** (Liu et al., 2022), a RoBERTa-base model adapted to political text, continually trained on 3 million news articles with a triplet loss objective. We further design **joint models** that are trained to predict both partisan events and article ideology. Finally, seeing an emerging research area of using large language models (LLMs), we further prompt **Chat-GPT** to detect events with a five-sentence context size. Appendix F contains an analysis of experiments with different context sizes and number of shots for prompting ChatGPT.

### 6.2 Results

For Partisan Event Detection task, we report macro F1 on each category of partisan events and on all categories in Table 2. For Ideology Prediction task, we use macro F1 score on both the left and the right ideologies, as reported in Table 3. First, both RoBERTa and POLITICS improve performance over the random baseline, where joint

|  | Partisan | Counter-Partisan | Neutral | Combined |
|---|---|---|---|---|
| Random | 24.74 | 7.83 | 67.50 | 33.36 |
| RoBERTa | 48.40 | 15.78 | 80.61 | 48.26 |
| +joint | 47.48 | 17.05 | 80.52 | 48.35 |
| POLITICS | **48.72** | **19.88** | 80.09 | **49.56** |
| +joint | 47.09 | 19.55 | **81.28** | 49.31 |
| ChatGPT | 31.42 | 9.99 | 71.75 | 37.62 |

Table 2: Model performance on partisan event detection task measured by macro F1 with the best results in **bold**. Average of 5 random seeds.

training yields further improvement for POLITICS on ideology prediction and a slight improvement for RoBERTa on event detection. Moreover, it is worth noting that partisan events are more easily detected than counter-partisan ones, which are usually implicitly signaled in the text and thus require more complex reasoning to uncover. Finally, though ChatGPT model has obtained impressive performance on many natural language understanding tasks, its performance is only better than a random baseline. This suggests that large language models still fall short of reasoning over political relations and ideology analysis that require the understanding of implicit sentiments and broader context.

### 6.3 Error Analysis

We further conduct an error analysis on event predictions by RoBERTa model. We discover that it fails to predict events with implicit sentiments and cannot distinguish the differences between partisan and counter-partisan events. To solve these two problems, future works may consider a broader context from the article, other articles on the same story, and the media itself, and leverage entity coreference and entity knowledge in general. More details on error analysis can be found at Appendix E.

## 7 Conclusion

We conducted a novel study on partisan and counter-partisan event reporting in news articles across ideologically varied media outlets. Our newly annotated dataset, PAC, illustrates clear partisan bias in event selection even among ostensibly mainstream news outlets, where counter-partisan event inclusion appears to be due to a combination of newsworthiness, market breadth, and emotional engagement. Experiments on partisan event detection with various models demonstrate the task's difficulty and that contextual information is important for models to understand media bias.

**Acknowledgments**

This work is supported in part by the National Science Foundation under grant III-2127747 and by the Air Force Office of Scientific Research through grant FA9550-22-1-0099. We would like to thank the anonymous reviewers for their helpful comments and feedback.

## 8 Limitations

Our study only focuses on American politics and the unidimensional left-right ideological spectrum, but other ideological differences may operate outside of this linear spectrum. Although our dataset already contains a diverse set of topics, other topics may become important in the future, and we will need to update our dataset. The conclusion we draw from the dataset may not be generalizable to other news media outlets. In the future work, we plan to apply our annotated dataset to infer events in a larger corpus of articles for better generalizability. The event detection model does not have perfect performance and may falsely classify biased content without any justifications, which can cause harm if people trust the model blindly. We encourage people to consider these aspects when using our dataset and models.

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

## A Annotation Guidelines

Below we include the full instructions for the annotators. A Javascript annotation interface is used to aid annotators during the process.

Given a pair of news articles, you need to first read two news articles and label their relative ideologies and absolute ideologies. Then, for each event, you need to follow these steps to label its partisanship (partisan, counter-partisan, or neural):

- Identify the agent entity and the patient entity for each event and other salient entities. These entities can be a political group, politicians, bills, legislation, political movements, or anything related to the topic of the article.

- Label each entity as left, neutral, or right based on the article context or additional information online.

- Estimate sentiments the author tries to convey toward each entity by reporting the events.

- Based on each entity, its ideology, and sentiments, you can decide whether an event supports or opposes the article's ideology. If it supports, label it as partisan. Otherwise, label it as counter-partisan. For example, in a right article, if a left entity is attacked or a right entity is praised by the author, you should label the event as a partisan event. If a left entity is praised or a right entity is attacked by the author, you should label the event as counter-partisan.

## B Annotation Quality

We collect stories from Allsides, a website presenting news stories from different media outlets. Its editorial team inspects news articles from different sources and pairs them together as a triplet. One of the authors, with sufficient background in American politics, manually inspected each story by following the steps below

- Read the summary from the Allsides, which includes the story context and comments from the editors on how each news article covers the story differently.

- Read each article carefully and compare them.

- Pick the two news articles with significant differences in their ideologies.

We hired six college students who major in political science, communication and media, and related fields to annotate our dataset. Three are native English speakers from the US, and the other three are international students with high English proficiency who have lived in the US for more than five years. All annotators were highly familiar with American politics. To further ensure the quality of the annotation, before the process began, we hosted a two-week training session, which required each annotator to complete pilot annotations for eight news articles and revise them based on feedback. After the training session, we held individual weekly meetings with each annotator to provide further personalized feedback and revise annotation guidelines if there was ambiguity. Each article is annotated by two students.

We calculate inter-annotator agreement (IAA) levels on the articles' relative ideologies, their absolute ideology, and the events. The IAA on the articles' relative ideologies between two annotators was 90%, while the agreement on the articles' absolute ideologies was 84%. The higher agreement on the articles' relative ideologies demonstrates the usefulness of treating a story as one unit for annotation. For stories with conflicting relative ideologies or articles with a difference greater than 1 in their absolute ideologies, a third annotator resolves all conflicts and corrects any mistakes. Despite the subjective nature of this task and the large number of events in each article, the Cohen's Kappa on event labels is 0.32, which indicates a fair agreement is achieved. When calculating agreement on whether a sentence contains a partisan or counter-partisan event when one exists, the score increases to 0.43, which is moderate agreement.

Our dataset covers diverse topics, including but not limited to immigration, abortion, guns, elections, healthcare, racism, energy, climate change, tax, federal budget, and LGBT.

## C Annotation Disagreement

In total, the dataset contains 304 news articles covering 152 news stories. All news stories are annotated by at least two annotators: 5 stories are annotated by one annotator and revised by another to add any missing labels and correct mistakes, while 147 stories are annotated by two annotators. Out of news stories annotated by two people, a third annotator manually merges 54 news articles to correct errors and resolve any conflicts. For the

| | P | R | F1 |
|---|---|---|---|
| Random | 51.71 | 51.72 | 51.54 |
| RoBERTa | 60.64 | 58.86 | 57.43 |
| +joint | 54.58 | 54.29 | 53.36 |
| POLITICS | **76.37** | 74.00 | 73.17 |
| +joint | 75.72 | **74.86** | **74.65** |

Table 3: Model performance on ideology prediction with the best results in **bold**. Average of 5 random seeds.

rest of the news stories, we combine annotations from two annotators and have a third annotator resolving only conflicting labels. During the merging process, we also discover three types of common annotation disagreements:

- Events with very weak intensity: some events are only annotated by one annotator, typically, these events have low intensity in their partisanship or are not relevant enough, so the other annotator skips them.

- Label different events within the same sentence: this happened the most frequently because when news articles report an event, they describe it with a cluster of smaller and related events. Two annotators may perceive differently which event(s) is partisan.

- Events are perceived differently by two annotators, one may think it is partisan, and the other may think it is counter-partisan. Usually, both interpretations are valid, and we have a third annotator to decide which interpretation should be kept.

## D    Ideology Prediction

Table 3 shows the ideology prediction performance of different models.

## E    Error Analysis

We perform a detailed examination of 100 event predictions generated by our RoBERTa model. We discover sentiments' intensity closely correlates with the model's performance. Specifically, when the model classifies events as either partisan or counter-partisan, 70% of these events feature strong/explicit event triggers like "opposing" or "deceived". The remaining events use more neutral triggers such as "said" or "passed". Our model demonstrates higher accuracy in predicting events that contain strong or explicit sentiments. However, it fails to predict events with implicit sentiments

and cannot distinguish the differences between partisan and counter-partisan events.

### E.1    Events with Implicit Sentiments

The first example in Figure 8 is from a news article about the climate emergency declared by Joe Biden after Congress failed the negotiation. The model fails to predict "give" as a partisan event. This is primarily because the term itself does not exhibit explicit sentiment and the model does not link "him" to Joe Biden. However, when contextualized within the broader scope of the article, it becomes evident that the author includes this event to bolster the argument for a climate emergency by highlighting its positive impact. To predict this type of events correctly, the model needs to understand the context surrounding the event and how each entity is portrayed and linked.

### E.2    Counter-partisan Events

The second example in Figure 8 is from a right news article about the lawsuit by Martha's Vineyard migrants against Ron DeSantis. The model incorrectly categorizes the event "horrified" as partisan due to the strong sentiment conveyed in the text. However, when placed in the broader context of the article, which defends Ron DeSantis and criticizes Democrats for politicizing migrants, this event should be more accurately classified as a counter-partisan event. The author includes it specifically to showcase the response from Democrats. The model seems to have limited capability of differentiating between partisan and counter-partisan events, possibly because of the similar language used to express partisan and counter-partisan events and the difficulty of recognizing the overall slant of news articles.

## F    ChatGPT Prompts

We use five different context sizes for our ChatGPT prompt: a story with two articles, a single article, 10 sentences, 5 sentences, and 3 sentences. An example prompt with sentences as context can be viewed in Table 5.

**Context Size vs. Number of Shots.**   Since the context window size of ChatGPT is fixed, we explore prompts with different context window sizes and investigate the trade-off between context window size and the number of shots. We try out five window sizes on our development set: 3 sentences, 5 sentences, 10 sentences, a single article,

| | | | | |
|---|---|---|---|---|
| **Article Title:** Biden ready to invoke 'domestic mobiliza-tion' against climate crisis after Congress failed | | | | |

**The Washington Times** (Left):
··· Declaring a climate emergency would **give** him the power to implement significant changes to energy produc-tion and consumption. ···

**Prediction**: Neutral
**Gold Label**: Partisan

**Article Title:** DeSantis Admin Slams Lawsuit Filed by Martha's Vineyard Migrants

**New York Post** (Right):
··· Newsom wrote on Thursday: Like millions of Amer-icans, I have been **horrified** at the images of migrants being shipped on buses and planes across the country to be used as political props. ····.

**Prediction**: Partisan
**Gold Label**: Counter-Partisan

Figure 8: Two errors made by the RoBERTa model.

| | Partisan | Counter-Partisan | Neutral | Combined |
|---|---|---|---|---|
| Story | 33.29 | 12.16 | 52.04 | 32.50 |
| + 1 shot | 34.51 | 8.42 | 54.89 | 32.61 |
| Article | 33.53 | 11.96 | 53.59 | 33.03 |
| + 2 shots | 34.66 | 10.48 | 57.53 | 34.29 |
| Sentence | | | | |
| 3 sentences | 20.95 | 11.05 | 70.54 | 34.18 |
| +34 shots | 28.67 | 3.57 | **79.33** | 37.19 |
| 5 sentences | 21.78 | **12.29** | 70.38 | 34.82 |
| +22 shots | 31.80 | 10.47 | 72.45 | **38.24** |
| 10 sentences | 27.99 | 12.10 | 66.06 | 35.38 |
| +11 shots | **35.71** | 10.74 | 64.17 | 36.87 |

Table 4: F1 scores of different ChatGPT prompts on the development set with the best results in **bold**. We use the maximum number of shots that can fit into each context window size. For each text input, we randomly sample demonstrations from the train set until the maximum number of shots can be fit. Average of 5 random seeds.

and a story with two articles of different ideolo-gies. As shown in Table 4, as the context window size increases, ChatGPT performs worse on neu-tral and counter-partisan events but improves its performance on partisan events. The larger context size gives ChatGPT more information about the article ideology, and its event detection results may be more biased toward the article ideology. We use the sentence prompt with 22 shots as our ChatGPT model for the test set.

| Prompt | Text |
| --- | --- |
| Sentence Prompt | Imagine you are a human annotator in a research study about news narratives. You will be asked to annotate partisan events and counter-partisan events in the news articles. "Partisan Events" further the interests, goals, and values of left or right. Authors reveal their ideology when including partisan events that benefit their ideology or foil their opposite ideology. "Counter-Partisan Events' are events that counter authors' ideology or support their opposite ideology to create a more memorable story or make articles seem unbiased. "Neutral Events" are events that authors include and present fairly and objectively. |
| | Given sentences from a news article, follow these instructions: 1. Read the sentences. 2. For each event <eid> in the article, perform the following tasks: a. Identify the agent entities, patient entities, and salient entities in the event. b. Identify the ideologies of all entities from step a. c. Identify the sentiment all entities from step a receive d. Identify whether the event is partisan, counter-partisan, or neutral based on its entities, ideologies, and sentiments from a, b, c. |
| | <Demostrations> |
| | <Sentences from a news article> |
| | Constraint: You should return your annotations in the following format without entities, without sentiments, without explanation: |
| | Partisan Events: eid, eid, eid, eid, ... |
| | Counter-Partisan Events: eid, eid, eid, eid, ... |
| | Neutral Events: eid, eid, eid, eid, ... |

Table 5: ChatGPT Prompt for sentences.