# OpenReview forum: "Crossing the Aisle: Unveiling Partisan and Counter-Partisan Events in News Reporting"
_EMNLP/2023/Conference — EMNLP 2023 Findings_

### Official Review · Reviewer_5vXQ · 2023-07-29

**Soundness:** 3

**Excitement:**

3: Ambivalent: It has merits (e.g., it reports state-of-the-art results, the idea is nice), but there are key weaknesses (e.g., it describes incremental work), and it can significantly benefit from another round of revision. However, I won't object to accepting it if my co-reviewers champion it.

**Paper Topic And Main Contributions:**

This paper proposes a news article dataset consisting of 206 news articles annotated for partisan and counter-partisan events in the articles. Partisan events are defined as events reporting of which benefit the stance of the news article and counter-partisan events are events that are reported with partisan events to make the news article seem neutral.

The authors proposed a human annotation framework for annotating such events by focusing on reported entities and sentiment toward them. The authors identified the ideologies of the news articles and the partisanship of the events using a few off-the-shelf classifiers and ChatGPT. The authors also presented some qualitative analysis of the partisan events and reported how different polarities use such events. For example, in the right-biased media, counter-partisan events appear towards the end of the news articles.

**Questions For The Authors:**

[Q1] What are the high-level discussion topics in the news articles studied in this paper?

**Reasons To Accept:**

[A1] This work proposes a novel dataset for analyzing partisan news media reporting of events.

[A2] The human annotation framework is well-described and it seems easy to replicate.

**Reasons To Reject:**

[R1] The concept of the partisan event as it is reported in the paper seems conceptually very similar to previously studied information bias (Fan et al., 2019) (discussed by the authors in the related works section) and subframes (Roy and Goldwasser, 2020) (not discussed by the authors in the paper). Hence, I am not totally convinced about the novelty of the work rather it seems like an incremental extension of the previous studies.

[R2] The number of news articles annotated is very few (only 206). Hence, it is hard to say that any trends found in this small amount of data are generalizable to the whole news media.

**Reproducibility:**

4: Could mostly reproduce the results, but there may be some variation because of sample variance or minor variations in their interpretation of the protocol or method.

**Reviewer Confidence:**

4: Quite sure. I tried to check the important points carefully. It's unlikely, though conceivable, that I missed something that should affect my ratings.

---

> ### Author Rebuttal · Authors · 2023-08-29
>
> Thank you for your valuable feedback and suggestions. We will address your questions in the revision.
>
> ***Re: R1***
>
> As outlined in our paper, our approach extends beyond the ***token-level*** spans of informational bias (Fan et al., 2019) to focus on ***event-level*** granularity. This allows for a more nuanced capture of the complex interactions among entities within news stories. In our revision, we intend to ***add comparisons against subframes*** (Roy and Goldwasser, 2020), which primarily concentrate on ***token-level*** bias owing to their reliance on ***manually curated lexicons*** for each topic. While their study covers only three topics, our dataset covers more than 10 controversial topics. It contains more extensive information (e.g., sentiments towards a participatory entity) that allows us to study how bias is subtly signaled in the news.
>
> Additionally, we introduce ***the novel concept of counter-partisan events***. This enables us to investigate not only the events that align with the media's stance but also those that run counter to it. By studying partisan and counter-partisan events collectively, we gain a more comprehensive understanding of media bias within news articles. Specifically, we emphasize the subtle ***content selection effect***, often overlooked by readers yet proven effective in persuasion (Groeling, 2013). Given the current computational limitations in studying this effect at scale, we are making our dataset publicly available to foster new insights into this critical area.
>
> ***Re: R2***
>
> As shown in Table 1, our dataset contains ***5,720 sentences, 4,612 annotations of partisan events, and 1,550 annotations of counter-partisan partisan events***. It contains ***more than 10 controversial political topics***. This comprehensive nature—boasting a high sentence count, rich annotations, and topic diversity—enables our dataset to offer a meaningful snapshot of news media trends. In comparison to similar datasets, BABE (Spinde et al., 2021) annotates 3,700 sentences; the sentence dataset (Lim et al., 2020) annotates 46 news articles, which contain 966 sentences; BASIL (Fan et al.,2019) consists of 300 news articles annotated with 1,727 spans. We are confident that our dataset is a more extensive resource for analyzing media bias.
>
> ***Re: Q1***
>
> Our dataset covers diverse topics, including ***immigration, abortion, guns, elections, healthcare, racism, national defense, climate change, federal budget, and Congress***. On average, each topic has about 20 news articles.
>
>
> We hope we have tackled all your concerns about our work and hope you can reconsider your evaluation. If there are any other issues, please let us know by updating your review, and we would be happy to resolve them in the revision.
>
> **References**
>
> [1] Lisa Fan, Marshall White, Eva Sharma, Ruisi Su, Prafulla Kumar Choubey, Ruihong Huang, and Lu Wang. 2019. In Plain Sight: Media Bias Through the Lens of Factual Reporting. In Proceedings of the 2019 Conference on Empirical Methods in Natural Language Processing and the 9th International Joint Conference on Natural Language Processing (EMNLP-IJCNLP), pages 6343–6349, Hong Kong, China. Association for Computational Linguistics.
>
> [2] Shamik Roy and Dan Goldwasser. 2020. Weakly Supervised Learning of Nuanced Frames for Analyzing Polarization in News Media. In Proceedings of the 2020 Conference on Empirical Methods in Natural Language Processing (EMNLP), pages 7698–7716, Online. Association for Computational Linguistics.
>
> [3] Groeling, T. (2013). Media Bias by the Numbers: Challenges and Opportunities in the Empirical Study of Partisan News. Annual Review of Political Science, 16(1), 129–151. https://doi.org/10.1146/annurev-polisci-040811-115123
>
> [4] Timo Spinde, Manuel Plank, Jan-David Krieger, Terry Ruas, Bela Gipp, and Akiko Aizawa. 2021. Neural Media Bias Detection Using Distant Supervision With BABE - Bias Annotations By Experts. In Findings of the Association for Computational Linguistics: EMNLP 2021, pages 1166–1177, Punta Cana, Dominican Republic. Association for Computational Linguistics.
>
> [5] Sora Lim, Adam Jatowt, Michael Färber, and Masatoshi Yoshikawa. 2020. Annotating and Analyzing Biased Sentences in News Articles using Crowdsourcing. In Proceedings of the Twelfth Language Resources and Evaluation Conference, pages 1478–1484, Marseille, France. European Language Resources Association.

---

### Official Review · Reviewer_wiNC · 2023-08-04

**Soundness:** 3

**Excitement:**

3: Ambivalent: It has merits (e.g., it reports state-of-the-art results, the idea is nice), but there are key weaknesses (e.g., it describes incremental work), and it can significantly benefit from another round of revision. However, I won't object to accepting it if my co-reviewers champion it.

**Paper Topic And Main Contributions:**

This work focuses on the task of detecting partisan vs counter-partisan events in news reporting. The authors aggregate 103 news stories, each with 2 news articles from opposing ideologies, end extract events from them using a RoBERTa-based model. They then annotate (a) each article as ideologically leaning towards left/right and on a Likert scale (1-5, left-right) and (b) each identified event in a news article as partisan/counter-partisan/neutral. After a qualitative analysis of their annotations, they test two RoBERTa-based models on two tasks: (A) article-level ideology prediction and (B) partisan event detection -- for (B) they also try ChatGPT.

The paper is well-written, albeit with a few clarification points needed primarily on the annotation process (§3). The major contribution is the introduction of a novel dataset, which can be useful for future research in this area. My major concern though is the lack of error and qualitative analysis of the results, which is quite important for this type of work: which are the most challenging cases for the best-performing model? Which cases are more easily captured? Is there a correlation between qualitative characteristics (e.g., sentiment) and the accuracy of the models? Incorporating a section on this type of analysis would greatly benefit the paper, as it can guide future work on this task/dataset.

**Questions For The Authors:**

- L134: "We manually inspect each story [...] differ": Please provide more details. Who inspected each story, which were the annotation guidelines/the inspectors' background, IAA, etc.

- L136: Which is the definition of "events from TimeML"?

- L138: How was the 89.31 F1 score calculated? On which test set? Is there a performance drop expected when applied on the PARTISAN EVENTS dataset?

- L174: "we held individual weekly meetings [...] if there was ambiguity": could you provide an example use case in an appendix possibly? Was this part of the training or part of the actual annotation task? If the latter is the case, could this introduce bias in the annotations?

- §3.2: Clarify early on this section that each annotation was performed by two students and not by all of them.

- §3.2: You mention that the IAA on "*stories'* relative ideology" is 91%; but on §3.1, you mention that "At the *article* level, the annotator determines the relative ideological ordering". Do these two quoted pieces of text refer to the same annotation? If so, please be consistent on your terminology (i.e., "story" vs "article level" annotation).

- L186: "a significant difference in their absolute ideologies": please provide the exact absolute value used.

- §5: Is Task A a two- or a five-class prediction task?

- Results section: it is unclear to my why you have omitted the neutral class. Surely the other two classes are more interesting to your task, but it is very important for the reader to see the overall picture of the results.

**Reasons To Accept:**

- The introduction of a novel dataset.

**Reasons To Reject:**

- Lack of error and qualitative analysis of the results, which can set up the future research directions on this task.
- A few missing details/clarification points are needed, primarily in section 3 (annotation process) which is the most important.

**Reproducibility:**

4: Could mostly reproduce the results, but there may be some variation because of sample variance or minor variations in their interpretation of the protocol or method.

**Reviewer Confidence:**

4: Quite sure. I tried to check the important points carefully. It's unlikely, though conceivable, that I missed something that should affect my ratings.

**Typos Grammar Style And Presentation Improvements:**

- L240: research vs reach?
- Table 3: add "." at the end
- L253: "hurts the model performance" for Task A

---

> ### Author Rebuttal · Authors · 2023-08-29
>
> Thank you for your valuable feedback and suggestions.
>
> ***Re: Lack of error and qualitative analysis***
>
> To address your concerns regarding error analysis, we performed ***a detailed examination of 100 event predictions*** generated by our top-performing model, the joint RoBERTa model. We discovered ***sentiments' intensity closely correlates with the model's performance***. Specifically, when the model classifies events as either partisan or counter-partisan, 70% of these events feature strong/explicit event triggers like "opposing" or "deceived." The remaining events use more neutral triggers such as "said" or "passed." Our model demonstrates higher accuracy in predicting events that contain strong or explicit sentiments.
>
>
> 1. Regarding errors made by the model, the first category of errors is ***the failure to detect events with implicit sentiments.***
> > Declaring a climate emergency would ***give*** him the power to implement significant changes to energy production and consumption.
>
>    The above sentence is from a news article about the climate emergency declared by Joe Biden after Congress failed the negotiation. The model fails to predict “give” as a partisan event. This is primarily because the term itself doesn't exhibit explicit sentiment and the model may not link "him" to Joe Biden. However, when contextualized within the broader scope of the article, it becomes evident that the author includes this event to bolster the argument for a climate emergency by highlighting its positive impact. To predict this type of events correctly, the model needs to ***understand the context surrounding the event and how each entity is portrayed and linked.***
>
> 2. The second category of errors by the model, which is also the most challenging case, is ***the failure to distinguish between partisan and counter-partisan events.***
> > Newsom wrote on Thursday: Like millions of Americans, I have been ***horrified*** at the images of migrants being shipped on buses and planes across the country to be used as political props.
>
>    The above sentence is from a right news article about the lawsuit by Martha's Vineyard migrants against Ron DeSantis. Our model incorrectly categorizes the event "horrified" as partisan due to the strong sentiment conveyed in the text. However, when placed in the broader context of the article, which defends Ron DeSantis and criticizes Democrats for politicizing migrants, this event should be more accurately classified as a counter-partisan event. The author includes it specifically to showcase the response from Democrats. The model seems to have limited capability of differentiating between partisan and counter-partisan events, possibly because of the similar language used to express partisan and counter-partisan events and the difficulty of recognizing the overall slant of news articles.
>
> To solve these two problems, future works may consider ***a broader context, use cross-document information, and utilize entity coreference and entity knowledge.*** We will include this error analysis in the camera-ready version.
>
> ***Re: Questions For The Authors***
>
> ***L134***: We collect stories from Allsides, a website presenting news stories from different media outlets. Its editorial team inspects news articles from different sources and pairs them together as a triplet. One of the authors, with sufficient background in American politics, manually inspected each story by
> 1. Read the summary from the Allsides, which includes the story context and comments from the editors on how each news article covers the story differently.
> 2. Read each article carefully and compare them
> 3. Pick the two news articles with significant differences in their ideologies
>
> ***L136***: In the TimeML annotation guidelines, events are defined as ***“a cover term for situations that happen or occur.”***
>
> ***L138***: We train a Roberta-Large model to detect event triggers with the training set from MATRES. Then, we evaluate the model on the test set from MATRES. The performance drop is ***minimal*** since both MATRES and the PARTISAN EVENTS dataset consist of news articles.
>
> ***L174***: One example of ambiguity is when the news article attacks a liberal politician for being too conservative on a specific issue. Although the left entity receive negative sentiment, the overall ideology of the news article is still left. The individual weekly meetings are held after the training. To ***minimize the bias***, we make the meeting mainly as a discussion to help annotators understand the annotation guidelines and clarify any confusion.
>
> ***3.2***: Each story is annotated by ***at least two people***. More details can be found in Appendix B.
>
> ***3.2***: Yes, they refer to the same annotation, which is the relative ideology. 91% of the news articles have the same relative ideology annotated by two annotators.
>
> ***L186***: The value is 2. For example, if a news article is annotated Left by one person and Center by the other, a third person will annotate the news article again to resolve conflicts and correct any mistakes.
>
> ***§5***: Task A is a two-class prediction task.
>
> ***Results section***: Omitting the neutral class was mainly to save space. We will include the performance of the neutral class in the camera-ready version.
>
> ***Presentation improvements***: We will incorporate them in the camera-ready version.
>
> We hope we have tackled all your concerns about our work and hope you can reconsider your evaluation. If there are any other issues, please let us know by updating your review, and we would be happy to resolve them in the revision.

---

### Official Review · Reviewer_ZkKA · 2023-08-05

**Soundness:** 2

**Excitement:**

2: Mediocre: This paper makes marginal contributions (vs non-contemporaneous work), so I would rather not see it in the conference.

**Paper Topic And Main Contributions:**

The paper studied the effects of partisan and counter-partisan events in news reporting across different media outlets. A newly annotated dataset (PARTISAN EVENTS) is provided. Experiments on partisan event detection with a variety of models demonstrate the difficulty of the proposed task.

**Reasons To Accept:**

(1) The proposed partisan and counter-partisan event detection task is new and interesting
(2) A task-corresponding dataset is built for research purpose
(3) The paper is well written and easy to follow


**Reasons To Reject:**

(1) The novelty of the paper is limited.
(2) It is mainly a case-study paper. No new methods and techniques are proposed. The experiments only show the difficulty of the proposed task.


**Reproducibility:**

3: Could reproduce the results with some difficulty. The settings of parameters are underspecified or subjectively determined; the training/evaluation data are not widely available.

**Reviewer Confidence:**

4: Quite sure. I tried to check the important points carefully. It's unlikely, though conceivable, that I missed something that should affect my ratings.

---

> ### Author Rebuttal · Authors · 2023-08-29
>
> Thank you for your time and feedback. We are glad you found our task “new and interesting”. We would like to highlight that our contributions are threefold.
>
> 1. We propose ***a new task of detecting partisan and counter-partisan events*** and provide multiple baselines of different model families, including fine-tuned PLM and recent LLM (ChatGPT), to show the difficulty of the proposed task. As we show in the paper, this task is extremely challenging due to ***the subtle language usage in partisan content and the media's tendency to obfuscate their biases***, especially for how events are selectively reported to promote certain media ideologies.
>
> 2. For this task, we collect and label ***the first dataset of 206 news articles with 6,162 partisan and counter-partisan events***, which span ***more than 10 representative and controversial topics*** such as abortion and gun control.
>
> 3. Our analyses of these models and data ***shed more light*** on the content selection effect and how partisan and counter-partisan events are used in the news by media representing different ideologies.
>
> To the best of our knowledge, we are the ***first*** to propose and define the task of detecting partisan and counter-partisan events.  Studying this task gives us a deeper understanding of media bias within news articles and how news articles select and organize content in the production process to sway readers’ opinions tacitly.
>
> While the models we evaluate are not new, as you mention, our experiments and analyses of multiple strong baselines on our novel task and dataset reveal ***the limitations of current computational tools for studying the content selection effect at scale*** and point to future work for better understanding this phenomenon. We are making our dataset available to the community as a resource that could help pave the way for new insights into this important issue.
>
> We hope we have tackled all your concerns about our work and hope you can reconsider your evaluation. If there are any other issues, please let us know by updating your review, and we would be happy to resolve them in the revision.

---

### Meta-Review · Area_Chair_qczK · 2023-09-19

**Recommendation:** 3

**Metareview:**

This paper introduces a newswire dataset with an annotation framework for labeling partisan and counter-partisan events.
Overall strengths include the novel dataset and annotation framework that are clearly presented. On the other hand, weaknesses include a lack of clear definition of partisan news, lack of error analysis or any insights on results, and the limited size of the dataset.

---

### Decision · Program_Chairs · 2023-10-07

**Decision:**

Accept-Findings

**Comment:**

This paper introduces a newswire dataset with an annotation framework for labeling partisan and counter-partisan events.
Overall strengths include the novel dataset and annotation framework that are clearly presented. On the other hand, weaknesses include a lack of clear definition of partisan news, lack of error analysis or any insights on results, and the limited size of the dataset.